# *KASPER*: Kolmogorov Arnold Networks for Stock Prediction and Explainable Regimes

**Vidhi Oad**[*]                                                   *vidhi.ec25@gmail.com*
*Vishwakarma Government Engineering College, Ahmedabad, India*

**Param Pathak**[*]                                               *param.pathak@fractal.ai*
*QuantumAI Lab, Fractal Analytics, Mumbai, India*

**Nouhaila Innan**                                               *nouhaila.innan@nyu.edu*
*eBRAIN Lab, Division of Engineering, New York University Abu Dhabi (NYUAD), Abu Dhabi, UAE*
*Center for Quantum and Topological Systems (CQTS), NYUAD Research Institute, NYUAD, Abu Dhabi, UAE*

**Shalini D**                                               *shalini.devendrababu@fractal.ai*
*QuantumAI Lab, Fractal Analytics, Gurugram, India*

**Muhammad Shafique**                                          *muhammad.shafique@nyu.edu*
*eBRAIN Lab, Division of Engineering, New York University Abu Dhabi (NYUAD), Abu Dhabi, UAE*
*Center for Quantum and Topological Systems (CQTS), NYUAD Research Institute, NYUAD, Abu Dhabi, UAE*

**Reviewed on OpenReview:** *https://openreview.net/forum?id=PD4jGJQtL8*

## Abstract

Forecasting in financial markets remains a significant challenge due to their nonlinear and regime-dependent dynamics. Traditional deep learning models, such as long short-term memory networks and multilayer perceptrons, often struggle to generalize across shifting market conditions, highlighting the need for a more adaptive and interpretable approach. To address this, we introduce Kolmogorov–Arnold networks for stock prediction and explainable regimes (*KASPER*), a novel framework that integrates regime detection, sparse spline-based function modeling, and symbolic rule extraction. The framework identifies hidden market conditions using a Gumbel-Softmax-based mechanism, enabling regime-specific forecasting. For each regime, it employs Kolmogorov–Arnold networks (KANs) with sparse spline activations to capture intricate price behaviors while maintaining robustness. Interpretability is achieved through symbolic learning based on Monte Carlo Shapley values, which extracts human-readable rules tailored to each regime. Applied to real-world financial time series from Yahoo Finance, the model achieves an $R^2$ score of 0.89, a Sharpe Ratio of 12.02, and a mean squared error as low as 0.0001, outperforming existing methods. This research establishes a new direction for regime-aware, transparent, and robust forecasting in financial markets.

## 1 Introduction

The stock market is a complex, dynamic system influenced by multiple factors, including economic factors, global events, and investor emotions. Predicting its behavior is a critical challenge with significant impact on financial decision-making, portfolio management, and risk assessment. However, the volatility, non-stationarity, and abrupt regime shifts in market behavior make accurate forecasting a challenging task (A & James, 2023).

The key problem we address is the accurate prediction of stock market behavior, particularly during regime shifts such as transitions between bullish, bearish, and stagnant phases. These shifts are important because

---

[*]These authors contributed equally.

they represent changes in market dynamics, and failing to adapt to them can lead to substantial financial losses (Kokare et al., 2022). Traditional models like Autoregressive Integrated Moving Average (ARIMA) (Ho & Xie, 1998), and Generalized Autoregressive Conditional Heteroskedasticity (GARCH) (Bauwens et al., 2006), while effective for stationary data, struggle to capture the nonlinearities and temporal dependencies inherent in financial time series (Alkhfajee & Al-Sultan, 2024; Crawford & Fratantoni, 2003).

State-of-the-art approaches to stock market prediction can be broadly categorized into statistical models, Machine Learning (ML) models, and hybrid methods, each with distinct strengths and limitations. Statistical models, such as Hidden Markov Models (HMMs) and regime-switching models (Yuan & Mitra, 2016), are interpretable and theoretically grounded but rely on rigid parametric assumptions, fixed transition probabilities, and Gaussian distributions, which fail to capture the nonlinearities and abrupt regime shifts of real-world markets (Wątorek et al., 2021), leading to poor performance during volatile periods. ML models, including Long Short-Term Memory networks (LSTMs), transformers, and Large Language Models (LLMs), excel at capturing complex patterns and long-term dependencies, with LLMs even incorporating external textual data for enhanced predictions.

However, their black-box nature limits interpretability (Chen et al., 2023), and they often struggle to adapt to distinct market regimes, either overfitting to historical patterns or introducing lookahead bias due to poor temporal alignment (Bhandari et al., 2022; Zhang et al., 2022). Hybrid methods aim to combine the strengths of both approaches, but they frequently fail to enforce sparsity, leading to overfitting (Islam et al., 2024). In this context, sparsity enforcement refers to applying L1 regularization ($\lambda\|W\|_1$) to the model parameters $W$ to drive many weights to zero, thereby reducing model complexity and mitigating overfitting. Most hybrid ensemble methods lack such explicit sparsity constraints on their feature-importance weights or ensemble coefficients. They also lack robust mechanisms for temporal alignment, which can result in data leakage and inflated performance metrics. More specifically, data leakage occurs when future information enters the training process, for example, by using future prices for normalization or allowing validation/test windows to overlap with training periods. This improper temporal separation inflates metrics such as accuracy, Sharpe ratio, and $R^2$ by allowing the model to learn from information it would not have access to at prediction time, leading to an overly optimistic assessment of real-world performance. Additionally, they often fail to provide clear, actionable insights into regime-specific drivers, limiting their practical utility (Haase & Neuenkirch, 2023). These limitations highlight the need for more advanced methods, particularly for modeling stock market behavior.

To address these challenges, we propose *KASPER*, a novel framework that uses KANs (Liu et al., 2024b) and integrates adaptive regime detection with sparse, interpretable feature engineering. Specifically, we enforce sparsity through L1 regularization on regime-specific forecast weights $w_j^{(i)}$ and dynamic feature masking, ensuring that only the most predictive signals influence each regime. This sparse modeling is implemented within KANs, which are inspired by the Kolmogorov-Arnold representation theorem (Schmidt-Hieber, 2021), stating that any multivariate continuous function can be decomposed into a finite sum of univariate functions. Accordingly, KANs implement this decomposition through learnable spline-based activation functions, enabling flexible function approximation. As illustrated in Fig. 1, the KAN architecture maps raw inputs through learnable spline-activated units (bottom squares) and then linearly recombines the transformed signals across successive layers, producing regime-aware predictions (top node). Each activation $\phi(x)$ is constructed from B-spline bases defined on a knot grid, i.e., a set of breakpoints $\{k_1, k_2, \ldots, k_m\}$ that partition the input domain into intervals, and is refined across progressively finer grid resolutions. The knot locations control the spline's local behavior by determining where basis functions are centered and how $\phi(x)$ transitions between polynomial segments as $x$ varies. This grid refinement improves expressive power while preserving interpretability, allowing KANs to represent high-dimensional mappings with fewer layers.

Our approach uses dynamic spline-based activations (Vecci et al., 1998) to capture nonlinear price dynamics and adapt to regime shifts through robust percentile-based initialization of spline knots. Specifically, for each feature dimension, knot positions are initialized based on the effective range of the training distribution. Mathematically, given a feature vector $\mathbf{x} \in \mathbb{R}^n$, we first determine the robust boundaries $x_{\min} = \text{quantile}(\mathbf{x}, p_{\min})$ and $x_{\max} = \text{quantile}(\mathbf{x}, p_{\max})$, where $p_{\min}$ and $p_{\max}$ are lower and upper percentiles (e.g., 0.01 and 0.99) chosen to exclude outliers. We then compute knot locations $\{k_g\}_{g=1}^{G}$ on a uniform grid within these bounds: $k_g = x_{\min} + (g-1)\frac{x_{\max}-x_{\min}}{G-1}$, where $G$ is the total number of grid points and $g$ indexes

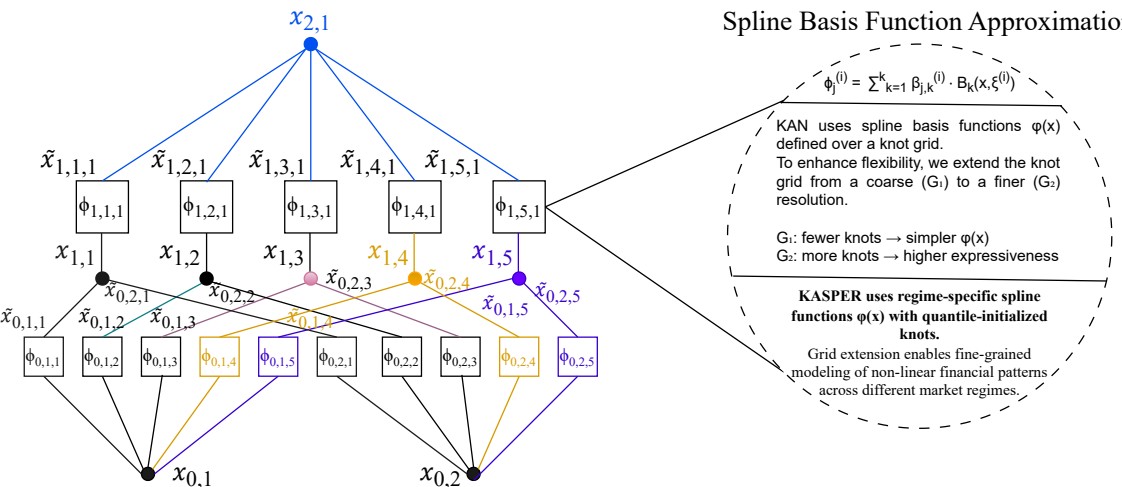

Figure 1: KAN architecture with two input features and spline-activated units across layers. Inputs $x_{0,1}$ and $x_{0,2}$ connect to spline units $\phi_{0,i,j}$ that produce transformed features $\tilde{x}0, i, j$, which are selectively routed to hidden nodes $x1, j$ and aggregated to the output $x_{2,1}$. Each $\tilde{x}_{l,i,j}$ denotes a learned univariate spline transformation applied to a linearly projected input, supporting interpretable feature-to-node mappings and sparse connectivity. The right inset illustrates spline basis-function approximation on a knot grid and the effect of grid extension $(G_1 \rightarrow G_2)$, where increased knot resolution improves expressiveness and enables smoother approximations.

the knot. This construction ensures that the spline bases cover the effective support of the distribution with consistent resolution and improved stability under extreme values. To ensure temporal alignment and prevent lookahead bias, we employ strict closed-left windows for rolling feature calculations and carefully shift historical data. To further improve generalization, sparsity is enforced through two complementary mechanisms. First, we apply L1 regularization to all model parameters, including spline basis coefficients and forecast weights. Second, we use dynamic masking that selectively zeros out small-magnitude spline coefficients $\beta_{j,k}^{(i)}$ based on regime-specific thresholds. This dual approach retains only the most significant weights and basis functions, reducing overfitting. For interpretability, we use Monte Carlo Shapley values (Ghorbani & Zou, 2019) to quantify the contribution of each feature to regime-specific predictions, enabling the extraction of actionable rules, such as identifying dominant features like High–Low (HL) in bearish regimes and Open–Close (OC) in bullish regimes. In particular, HL is defined as $\mathrm{HL}_t = \ln\left(\frac{H_t}{L_t}\right)$, where $H_t$ and $L_t$ are the daily high and low prices, and OC is defined as $\mathrm{OC}_t = \ln\left(\frac{C_t}{O_t}\right)$, where $O_t$ and $C_t$ are the opening and closing prices. This combination of adaptive modeling, sparsity enforcement, and interpretability enables regime-specific analysis across diverse market conditions.

**The key contributions of this study are as follows:**

- A novel Regime-Adaptive Forecasting Layer that makes use of sparsely activated splines to model distinct market dynamics across regimes. This mitigates the overfitting risk in financial time series by ensuring sparsity-constrained representations that generalize well.

- An orthogonality constraint within the regime detection network to enforce disentangled representations for different market states. This prevents feature collapse, ensuring each regime retains distinct and interpretable characteristics.

- For seamless discernibility of the regimes, Contrastive Regularization is used. It helps maximize inter-regime dissimilarity while maintaining intra-regime coherence, thereby improving the stability of regime classification and reducing misclassification in volatile market conditions.

- Utilizing a Monte Carlo Shapley method with temporal weighting to extract interpretable, regime-specific rules. This enhances the model's transparency by identifying dominant factors influencing each market regime.

The rest of the paper is organized as follows: Sec. 2 reviews related work on market regime detection, KAN-based financial modeling, and interpretability in finance; Sec. 3 presents the proposed *KASPER* framework; Sec. 4 reports the experimental setup and results, including ablation studies and comparisons with baseline models; finally, Sec. 5 concludes with a summary of the main findings and directions for future work.

## 2 Background and Related Work

Recent work in financial forecasting includes hybrid deep architectures for price prediction (Liu et al., 2025; Elhoseny et al., 2025), transformer-based models for capturing long-range temporal dependencies under volatility (Kabir et al., 2025; Hadizadeh et al., 2025), and reinforcement-learning frameworks for portfolio decision-making and risk control in non-stationary markets (Yao, 2025; Ye et al., 2020). Consistent with the scope of *KASPER*, this section focuses on market regime detection, KAN-based approaches for financial time series, and explainability/interpretability methods in finance.

### 2.1 Market Regime Detection

Financial markets transition between distinct regimes, characterized by varying volatility, return distributions, and risk factors. Early work established regime switching via Markov models (Hamilton, 1989):

$$r_t = \mu_{z_t} + \epsilon_t, \quad \epsilon_t \sim \mathcal{N}(0, \sigma^2_{z_t}), \tag{1}$$

$$P(z_t = j \mid z_{t-1} = i) = A_{ij}. \tag{2}$$

where $r_t$ is the observed return at time $t$, $z_t$ is the latent regime indicator, and $A$ is a fixed transition matrix. While interpretable, this approach suffers from static regime centroids and rigid parameters.

Ang and Bekaert (Ang & Bekaert, 2002), showed that modeling asset returns as regime-dependent improves forecasting and risk management:

$$r_t \mid z_t = k \sim \mathcal{N}(\mu_k, \sigma^2_k), \tag{3}$$

with portfolio weights dynamically adjusted according to the inferred regime.

Building on this, Guidolin and Timmermann (Guidolin & Timmermann, 2007) extended the framework using a multivariate Markov-switching model with four regimes (crash, slow growth, bull, and recovery). Regime transitions follow:

$$P(z_t = i \mid z_{t-1} = j) = A_{ji}, \quad i, j = 1, \dots, k, \tag{4}$$

where $A_{ji}$ is the transition probability. In their portfolio choice setting, the investor's dynamic optimization problem at time $t_b$ is formulated as:

$$J(W_b, r_b, z_b, \theta_b, \pi_b, t_b) = \max_{\boldsymbol{\omega}} \mathbb{E}_{t_b} \left[ \frac{W_B^{1-\gamma}}{1-\gamma} \right], \tag{5}$$

where $W_b$ denotes the investor's wealth at time $t_b$, $r_b$ and $z_b$ are the vectors of historical asset returns and predictor variables, respectively, $\theta_b$ collects all parameters of the regime-switching process, $\pi_b$ is the vector of filtered regime probabilities at $t_b$, and $t_b$ is a discrete decision epoch (rebalancing date) with $b = 0, 1, \dots, B-1$. The maximization is performed over the vector of portfolio weights $\boldsymbol{\omega}$ allocated to risky assets, and the objective is the expected utility of terminal wealth $W_B$ with relative risk aversion $\gamma$, conditional on information available at $t_b$. Bayesian updating of regime probabilities is performed via:

$$\pi_{b+1}(\theta_t) = \frac{(\pi_b^0(\theta_t)P_t^\varphi) \odot \eta(y_{b+1}; \theta_t)}{[(\pi_b^0(\theta_t)P_t^\varphi) \odot \eta(y_{b+1}; \theta_t)]^0 \iota_k}. \tag{6}$$

The Vector Smooth Transition Autoregressive (VLSTAR) model (Bucci & Ciciretti, 2021) was also an alternative that used a continuous transition function:

$$y_t = \mu_0 + \sum_{j=1}^{p} \varphi_{0,j} y_{t-j} + A_0 x_t + G_t(s_t; \gamma, c) \left[ \mu_1 + \sum_{j=1}^{p} \varphi_{1,j} y_{t-j} + A_1 x_t \right] + \varepsilon_t, \tag{7}$$

with the logistic function defined by:

$$G_t(s_t; \gamma, c) = \left[ 1 + \exp(-\gamma(s_t - c)) \right]^{-1}. \tag{8}$$

where $s_t$ triggers regime shifts and $\gamma$ controls transition smoothness (low $\gamma$ yields gradual changes; high $\gamma$ produces abrupt shifts). This model dynamically captures volatility shifts in market conditions.

## 2.2 KANs for Finance

KANs are effective in financial modeling, particularly for option pricing and stock prediction. Existing Finance-Informed Neural Networks (FINNs) for option pricing suffer from poor learning efficiency, high computational costs, complex training processes, and limited interpretability due to intricate loss functions requiring larger architectures and extensive training to maintain accuracy. To address these challenges, Liu et al. (Liu et al., 2024a) proposed an FINN *KAFIN*, that integrated financial equations into a neural framework. The Black–Scholes formula (Barles & Soner, 1998), which governed their pricing model can be represented as:

$$\frac{\partial C}{\partial t} + \frac{1}{2} \sigma^2 S^2 \frac{\partial^2 C}{\partial S^2} + rS \frac{\partial C}{\partial S} - rC = 0, \tag{9}$$

where $C(S, t)$ is the price of a financial option, $S$ is the price of the underlying asset, $\sigma$ is the volatility of the asset's returns, and $r$ is the risk-free interest rate. The price function is approximated using:

$$C(S, t) = \sum_{i=1}^{N} g_i(h_i(S), t), \tag{10}$$

where $g_i$ and $h_i(x)$ are trainable functions. The model minimizes a loss function incorporating financial constraints:

$$L(\theta) = \lambda_{\text{init}} L_{\text{init}} + \lambda_{\text{boundary}} L_{\text{boundary}} + \lambda_{\text{financial}} L_{\text{financial}}. \tag{11}$$

On the other hand, to enhance stock price prediction, Yao (Yao, 2024), proposed an LSTM-KAN hybrid model, where LSTM captures temporal dependencies and KAN refines nonlinear patterns. KAN improves nonlinear mapping via:

$$f(x) = \sum_{i=1}^{2n+1} g_i \left( \sum_{j=1}^{n} h_{ij}(x_j) \right), \tag{12}$$

where $g_i$ and $h_{ij}$ are trainable functions the model is trained using. Empirical results showed that KAFIN enhanced option pricing accuracy, while the LSTM-KAN model significantly reduced stock forecasting errors.

## 2.3 Explainability and Interpretability in Finance

Explainable AI (XAI) techniques address the transparency challenge in black-box deep reinforcement learning (DRL) models for portfolio management, where deep NNs provide limited insight into their decision-making. De-la-Rica-Escudero *et al.* (de-la Rica-Escudero et al., 2025) noted that many DRL approaches provide explanations only during training and do not support monitoring the agent's behavior at trading time; only a small number of prior studies explicitly considered explainability. They proposed a post hoc Explainable DRL framework that combines Proximal Policy Optimization with model-agnostic XAI methods (feature importance, SHAP, and LIME) by saving state–action pairs during training and generating explanations at inference time. Evaluated on five U.S. technology stocks using daily OHLC data (2015–2018), the framework highlighted asset-specific variations in OHLC importance and produced instance-level explanations of portfolio

allocations, with statistical validation of Shapley-value stability (variance below $10^{-7}$, $p < 0.01$ across 50 repetitions), enabling real-time monitoring of agent decisions during trading.

Interpretability challenges also persist in financial AI systems where complex architectures obscure the factors driving predictions. Ni *et al.* (Ni et al., 2025) argued that standard time-series visualization is often insufficient to reveal the temporal patterns underlying model decisions in financial risk assessment, which can reduce analyst trust and complicate regulatory compliance. They introduced a contrastive visual analytics framework that combines entropy-based temporal importance weighting with interactive dimensionality reduction (t-SNE, UMAP) for multivariate financial time series, enabling direct comparison of normal versus anomalous patterns and highlighting differences in feature attributions across risk scenarios. Empirical evaluation on credit risk assessment (5,248 corporate cases) and market volatility prediction reported reduced analyst decision time and improved agreement and anomaly detection, while maintaining sub-100 ms response times for interactive exploration.

Despite these advances, important gaps remain. Traditional regime detection methods rely on fixed transition matrices that require manual specification, while many recent Deep Learning (DL) models apply uniform processing without adapting to distinct market conditions. Hybrid architectures can achieve strong predictive accuracy, but they often lack explicit regime-switching mechanisms and provide primarily global explanations rather than regime-specific interpretability. Our work addresses these gaps by jointly performing regime detection and forecasting via differentiable Gumbel–Softmax classification, introducing Monte Carlo Shapley values with temporal weighting for regime-specific interpretability, and enforcing regime-specific sparsity through orthogonality constraints to mitigate feature collapse.

## 3 *KASPER* Framework

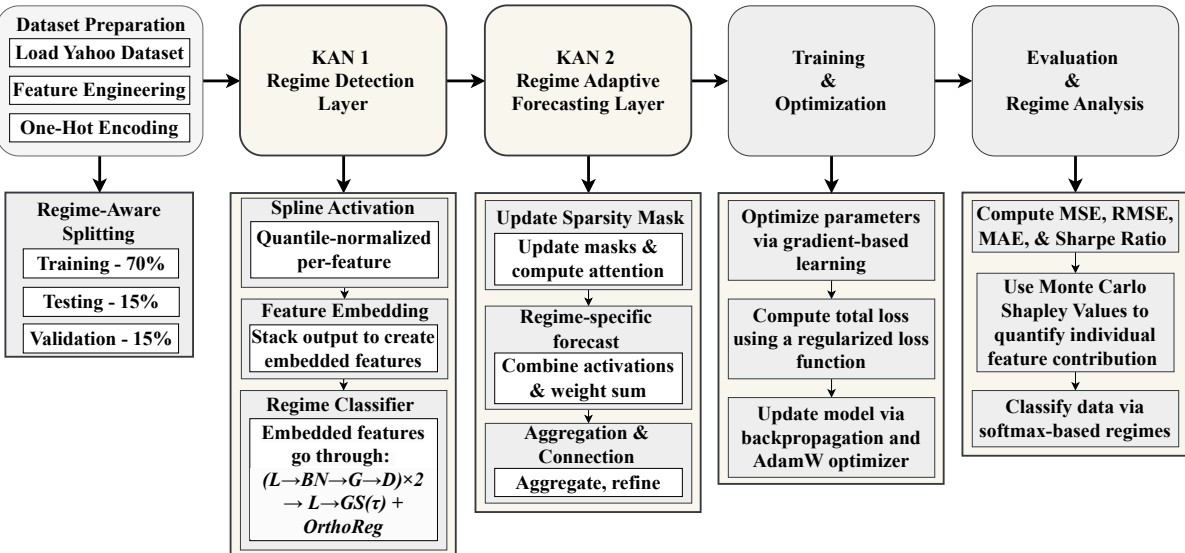

Figure 2: Workflow of the KASPER framework. The pipeline begins with Yahoo dataset preparation, feature engineering, one-hot encoding, and regime-aware splitting into training (70%), validation (15%), and testing (15%). KAN-1 performs regime detection using quantile-normalized spline activations, learns embedded features, and assigns regimes via a Gumbel-Softmax classifier with an orthogonality regularizer. KAN-2 produces regime-adaptive forecasts by updating a sparsity mask over spline components and computing attention-based aggregation before the final refinement. Parameters are learned with gradient-based optimization (AdamW) under a regularized composite loss. Performance is evaluated using MSE, RMSE, MAE, and Sharpe Ratio, while interpretability is provided via Monte Carlo Shapley value estimation and regime-specific rule extraction.

The proposed *KASPER* framework is composed of two core stages: regime detection and regime-adaptive forecasting, as illustrated in Fig. 2 and described in Algorithm 1. The input consists of an $n$-day window of financial time series data, structured as a state matrix $\mathbf{\Phi}_t \in \mathbb{R}^{n \times f}$, where $n$ denotes the number of historical

days and $f$ the number of features per day. Each row in $\boldsymbol{\Phi}_t$ captures log-transformed returns and volatility indicators to stabilize variance and enhance stationarity:

$$
\begin{aligned}
\boldsymbol{\Phi}_{t,i} = \bigg[ &\ln\left(\frac{p_{t-i+1}}{p_{t-i}}\right), \ln\left(\frac{f_{t-i+1}^{\text{High}}}{f_{t-i}^{\text{High}}}\right), \ln\left(\frac{f_{t-i+1}^{\text{Low}}}{f_{t-i}^{\text{Low}}}\right), \ln\left(\frac{f_{t-i+1}^{\text{Open}}}{f_{t-i}^{\text{Open}}}\right), \\
&\ln\left(\frac{V_{t-i+1}}{V_{t-i}}\right), \text{HL\_spread}_{t-i}, \text{OC\_spread}_{t-i}, \text{ATR}_{t-i}, \text{volatility\_ratio}_{t-i}, \\
&\text{price\_velocity}_{t-i}, \text{price\_acceleration}_{t-i}, \text{volume\_change}_{t-i}, \text{volume\_state}_{t-i} \bigg],
\end{aligned}
\tag{13}
$$

where $p_{t-i}$ denotes the closing price and $f_{t-i}^{\text{High}}$ and $f_{t-i}^{\text{Low}}$ denote the high and low prices, respectively, on day $t-i$. The engineered features include HL (denoted as HL\_spread), OC (denoted as OC\_spread), Average True Range (ATR), volatility ratio, price velocity, price acceleration, volume change, and volume state. These eight features are selected via `SelectKBest` with *f\_regression* scoring and form the final input to *KASPER*; further details are provided in Sec. 4.1.

---

**Algorithm 1:** KASPER

---

**Input:** Raw financial dataset $\mathcal{D}$
**Output:** Trained model, regime-specific rules, performance metrics

`// Step 1: Preprocessing and Feature Engineering`

**1** Load dataset $\mathcal{D}$ and forward-fill missing values
**2** Extract features: lags, rolling statistics, volatility, volume, momentum, temporal dummies
**3** Compute target as: $y_t = \frac{C_{t+1} - C_t}{C_t}$
**4** Split data into $\mathcal{D}_{train}, \mathcal{D}_{val}, \mathcal{D}_{test}$
**5** Apply feature selection via `SelectKBest`
**6** Standardize features using `StandardScaler`

`// Step 2: Define KASPER Architecture`

**7** Define Spline Activation for nonlinear mapping
**8** Create Regime Detection Layer to estimate soft regime probabilities using Gumbel-Softmax
**9** Initialize Regime Adaptive Forecasting Layer with regime-specific basis functions and attention mechanism
**10** Combine both layers into `KASPER` model

`// Step 3: Train the Model`

**11** **for** *epoch = 1 to N* **do**
**12**      **for** *each batch $(x_i, y_i) \in \mathcal{D}_{train}$* **do**
**13**          Compute predictions $\hat{y}_i$, regime probabilities $p_i$, embeddings $z_i$
**14**          Compute loss: $\mathcal{L} = \mathcal{L}_{pred} + \lambda_c \mathcal{L}_{contrastive} + \lambda_s \mathcal{L}_{sparse} + \lambda_o \mathcal{L}_{orthogonal} + \lambda_b \mathcal{L}_{balance}$
**15**          Backpropagate and update parameters
**16**      Evaluate on $\mathcal{D}_{val}$ and apply early stopping if needed

`// Step 4: Extract Regime-Specific Rules`

**17** **for** *regime $r = 1$ to $R$* **do**
**18**      Identify samples with highest $p_r$ from regime probabilities
**19**      Compute Shapley value $\phi_i^{(r)}$ for each feature $i$
**20**      Select top-3 contributing features as rules for regime $r$

`// Step 5: Evaluate Financial Metrics`

**21** Apply model to $\mathcal{D}_{test}$ to compute:
**22**      Sharpe Ratio, Max Drawdown, Cumulative Returns, Win Rate, Direction Accuracy

---

### 3.1 KAN Layer 1: Regime Detection

The regime detection module is designed to uncover latent market regimes using spline-activated KANs. This layer incorporates four components: spline activation functions, Gumbel-Softmax-based regime classification, contrastive loss for representation separation, and orthogonality regularization to enforce disentangled regime-specific embeddings.

### 3.1.1 Spline Activation Function

Each input feature is processed through a hybrid spline activation function that captures both linear and nonlinear trends:

$$f(x) = L(x) + C(x), \tag{14}$$

where $L(x)$ and $C(x)$ are the linear and cubic components, respectively. These are defined as:

$$L(x) = \sum_{m=0}^{N_{\text{linear}}-1} \tanh(w_m) \left[ \text{ReLU}(x_{\text{norm}} - k_m) - \text{ReLU}(x_{\text{norm}} - k_{m+1}) \right], \tag{15}$$

$$C(x) = \sum_{m=0}^{N_{\text{cubic}}-1} \sigma(v_m) x_{\text{norm}}^3, \tag{16}$$

where $w_m$ and $v_m$ are trainable parameters, $\sigma(\cdot)$ denotes the sigmoid function, and $x_{\text{norm}}$ is the input normalized with respect to the knot sequence $\{k_m\}$. To improve robustness to outliers, we place the knots $\{k_m\}_{m=1}^G$ on a uniform grid bounded by empirical percentiles of the input feature $\mathbf{x}$. Specifically, the knots are computed as $k_m = x_{\text{min}} + (m-1)\frac{x_{\text{max}} - x_{\text{min}}}{G-1}$, where $x_{\text{min}} = \text{quantile}(\mathbf{x}, p_{\text{min}})$ and $x_{\text{max}} = \text{quantile}(\mathbf{x}, p_{\text{max}})$ are the lower and upper percentile bounds (e.g., $p_{\text{min}} = 0.01$ and $p_{\text{max}} = 0.99$), and $G$ is the grid size.

### 3.1.2 Differentiable Regime Classification via Gumbel-Softmax

The classification of regime probabilities is achieved through the Gumbel-Softmax function, which provides a differentiable approximation to categorical sampling. We denote regimes by the index $r \in \{1, \dots, R\}$, where $R$ is the total number of regimes (set to 3 in our experiments):

$$P_t^{(r)} = \frac{\exp\left(f_r(\mathbf{\Phi}_t + g_r)/\tau\right)}{\sum_{s=1}^{R} \exp\left(f_s(\mathbf{\Phi}_t + g_s)/\tau\right)}, \tag{17}$$

where $\mathbf{\Phi}_t$ represents the input feature matrix at time $t$, $f_r$ denotes the spline-transformed activation corresponding to regime $r$, $g_r$ is the Gumbel noise sampled from $\text{Gumbel}(0, 1)$, $\tau$ is the temperature parameter, and $R$ is the total number of regimes.

### 3.1.3 Contrastive Loss for Regime Separation

To ensure that the latent embeddings corresponding to the same regime remain well clustered, a contrastive loss is introduced:

$$\mathcal{L}_{\text{contrastive}} = \mathbb{E}\left[\|z_i - z_j\|^2 \cdot y_{ij}\right], \tag{18}$$

where $z_i$ and $z_j$ denote the latent embeddings of samples $i$ and $j$, and $y_{ij} \in \{0, 1\}$ is a binary indicator defined as $y_{ij} = 1$ if samples $i$ and $j$ belong to the same regime and $y_{ij} = 0$ otherwise. Specifically, $z_i = \Phi_{\text{detect}}(\mathbf{x}_i)$ is the output of the spline-based regime detection layer for input $\mathbf{x}_i \in \mathbb{R}^n$, before the softmax classification step. The expectation $\mathbb{E}$ is taken over the empirical distribution of the training mini-batch $\mathcal{B}$, with inputs $\mathbf{x}$ treated as random samples from the market data distribution $\mathcal{D}$. This term enforces intra-regime compactness by minimizing the squared distance between embeddings assigned to the same regime.

### 3.1.4 Orthogonality Regularization

To enforce distinctiveness across regime-specific transformations, orthogonality is imposed on the regime weight vectors:

$$\mathcal{L}_{\text{orth}} = \|WW^T - I_R\|_F^2, \tag{19}$$

where $W \in \mathbb{R}^{R \times F}$ is the global weight matrix formed by vertically stacking the regime-specific row vectors $W_r$ for all $R$ regimes, and $I_R$ is the $R \times R$ identity matrix. Specifically, the $r$-th row of $W$ is $W_r = \left[w_1^{(r)}, w_2^{(r)}, \dots, w_F^{(r)}\right]$, where $w_j^{(r)}$ is the learnable scalar weight for the $j$-th feature in regime $r$, and $F$ is the total number of features.

### 3.2 KAN Layer 2: Regime-Adaptive Forecasting

Following regime identification, forecasting is performed using a second KAN layer with regime-specific parameterization. This layer employs sparse spline basis functions and is optimized using a composite objective function incorporating robustness, sparsity, and structural regularization.

#### 3.2.1 Forecasting Model

The predicted return $\hat{y}_t^{(r)}$ for regime $r$ is defined as:

$$\hat{y}_t^{(r)} = \sum_{j=1}^{F} w_j^{(r)} \phi_j^{(r)}(\mathbf{\Phi}_t), \tag{20}$$

where $\mathbf{\Phi}_t$ is the feature matrix, $\phi_j^{(r)}$ are spline basis functions and $w_j^{(r)}$ are trainable coefficients.

#### 3.2.2 Spline Basis Functions

Each basis function $\phi_j^{(r)}$ is constructed using regime-specific B-splines:

$$\phi_j^{(r)}(\mathbf{\Phi}_t) = \sum_{k=1}^{K} \beta_{j,k}^{(r)} B_k(\mathbf{\Phi}_t; \xi^{(r)}), \tag{21}$$

where $B_k$ denotes the $k$-th B-spline basis function with knots $\xi^{(r)}$, and $\beta_{j,k}^{(r)}$ are spline coefficients learned from data.

#### 3.2.3 Sparsity Enforcement

To promote interpretability and reduce overfitting, an $\ell_1$-regularization term is applied:

$$w_j^{(r)} \propto \mathrm{ReLU}(|w_j^{(r)}| - \theta^{(r)}), \tag{22}$$

where $\theta^{(r)}$ denotes a regime-specific sparsity threshold. The sparsity level is adaptively controlled through validation-based tuning of $\lambda$.

#### 3.2.4 Composite Loss Function

The full objective function for training integrates multiple loss components:

$$\mathcal{L} = \mathcal{L}_{\mathrm{Huber}} + \lambda_s \sum_{p \in \Theta} |p| + \lambda_c \mathcal{L}_{\mathrm{contrastive}} + \lambda_o \mathcal{L}_{\mathrm{orth}} + \lambda_b \mathcal{L}_{\mathrm{balance}}, \tag{23}$$

with $\Theta$ representing the set of all trainable model parameters, $|p|$ denotes the absolute value (L1 norm) of each individual parameter $p$. This composite formulation ensures: (1) robustness to outliers through Huber loss, (2) regime disentanglement via contrastive loss, (3) model sparsity through L1 regularization, (4) distinct regime representations via orthogonality constraints, and (5) balanced regime distribution across training samples.

### 3.3 Interpretability through Shapley-Based Rule Extraction

To enhance transparency, *KASPER* employs a Shapley value-based approach to interpret regime-specific forecasts.

### 3.3.1  Shapley Value Estimation

For a given feature $j$ in regime $r$, its contribution is quantified as:

$$\phi_j^{(r)} = \sum_{S \subseteq F \setminus \{j\}} \frac{|S|!(|F| - |S| - 1)!}{|F|!} \times [f^{(r)}(S \cup \{j\}) - f^{(r)}(S)], \tag{24}$$

where $F$ denotes the full feature set, and $f^{(r)}(S)$ is the regime-$r$ model output with subset $S$.

### 3.3.2  Monte Carlo Approximation

To approximate Shapley values efficiently, Monte Carlo sampling is used:

$$\hat{\phi}_j^{(r)} = \frac{1}{M} \sum_{m=1}^{M} [f^{(r)}(S_m \cup \{j\}) - f^{(r)}(S_m)], \tag{25}$$

where $S_m$ denotes randomly selected feature subsets (coalitions) and $M$ is the number of Monte Carlo samples.

### 3.3.3  Temporal Weighting Scheme

To emphasize recent market behavior, temporal weighting is applied to the sequence of past Shapley values:

$$\tilde{\phi}_j^{(r)} = \sum_{t=1}^{T} w_t \phi_j^{(r),t}, \quad w_t = \frac{\gamma^{T-t}}{\sum_{t=1}^{T} \gamma^{T-t}}, \tag{26}$$

where $\gamma \in (0,1)$ is the decay factor controlling the emphasis on recent time steps.

### 3.3.4  Regime-Specific Rule Extraction

For each regime $r$, the top three most influential features are selected:

$$\mathcal{R}_r = \left\{ \arg\max_{j \in F} \left| \tilde{\phi}_j^{(r)} \right| \,\middle|\, j = 1, 2, 3 \right\}, \forall r \in \{1, \ldots, R\}, \tag{27}$$

resulting in interpretable rules of the form:

$$\text{Regime } r: \quad X_{j_1} + X_{j_2} + X_{j_3} \rightarrow Y_r, \tag{28}$$

where $X_{j_1}, X_{j_2}, X_{j_3}$ are the dominant features and $Y_r$ denotes the forecasted market response under regime $r$.

## 4  Results and Discussion

### 4.1  Experimental Setup

We evaluate *KASPER* on the *Yahoo Finance* dataset (ARORA, 2023) for regime-aware stock return forecasting. The full experimental configuration is summarized in Table 1. To ensure a leakage-free setting, missing values are forward-filled and all rolling features are computed using strict closed-left windows, so that only information available up to time $t$ is used to construct the inputs at $t$.

Under this setup, *KASPER* is instantiated as a two-layer KAN model with two main components: a *RegimeDetectionLayer*, which learns spline-based representations and infers market states via Gumbel–Softmax classification, and a *RegimeAdaptiveForecastingLayer*, which produces regime-conditional forecasts. Orthogonality regularization promotes distinct regime representations, while the contrastive objective increases inter-regime separability, improving the stability of regime assignments.

Given the emphasis on temporal consistency, we engineer the input features using only historical observations to capture price dynamics, volatility structure, volume behavior, and regime context, with all rolling

Table 1: Experimental configuration of KASPER, including dataset details, preprocessing, model architecture, training setup, and evaluation metrics.

| Category | Parameter | Value/Description |
|---|---|---|
| Dataset | Source | *Yahoo Finance Dataset* |
| | Time Range | 2018–2023 |
| Data Preprocessing | Splitting Strategy | 70%–15%–15% temporal train-val-test split |
| | Feature Engineering | 15 engineered features (lags, rolling statistics, volatility, price dynamics) |
| | Feature Selection | 8 features via SelectKBest with *f_regression* |
| | Scaling Method | StandardScaler for both features and target |
| Model Architecture | Number of Regimes | 3 |
| | Hidden Dimension | 64 |
| | Spline Configuration | Hybrid linear (3) and cubic (2) splines with quantile-based initialization |
| | Activation Functions | SplineActivation and GELU |
| Training Parameters | Optimizer | AdamW (lr=0.001, weight_decay=1e-5) |
| | Batch Size | 32 |
| | Loss Function | Huber loss with composite regularization |
| | Regularization Weights | Contrastive: 0.01, Sparsity: 0.001, Orthogonality: 0.01, Regime Balance: 0.05 |
| | Epochs | 100 with early stopping (patience=15) |
| | Learning Rate Scheduler | ReduceLROnPlateau (factor=0.7, patience=7) |
| | Gradient Clipping | 0.5 |
| Evaluation Metrics | Statistical | R, MAE, $R^2$ |
| | Financial | Sharpe Ratio, Direction Accuracy, Max Drawdown, Win Rate, Profit Factor |

computations respecting the closed-left window constraint. The feature set includes HL and OC spreads, $k$-day log-returns $r_t^{(k)} = \ln(C_t/C_{t-k})$ for $k \in \{1, 7\}$, and a 21-day rolling volatility computed on past returns only: $\bar{r}_t = \frac{1}{w}\sum_{i=1}^{w} r_{t-i}^{(1)}$ and $\sigma_t = \sqrt{\frac{1}{w}\sum_{i=1}^{w}\left(r_{t-i}^{(1)} - \bar{r}_t\right)^2}$ with $w = 21$. We further compute the volatility ratio $\mathrm{VR}_t = \sigma_t/(|\bar{r}_t| + \varepsilon)$ and the Average True Range $\mathrm{ATR}_t = \frac{1}{w}\sum_{i=1}^{w} \max\left(H_{t-i} - L_{t-i}, |L_{t-i} - C_{t-i-1}|, |L_{t-i} - C_{t-i-1}|\right)$ to capture gap-aware price variability. Price momentum is summarized via velocity $U_t = C_t - C_{t-1}$ and acceleration $A_t = U_t - U_{t-1}$. Volume dynamics are represented using the normalized volume change $\Delta \mathrm{Vol}_t = (\mathrm{Vol}_t - \overline{\mathrm{Vol}}_t)/(\sigma_{\mathrm{Vol},t} + \varepsilon)$, where $\overline{\mathrm{Vol}}_t = \frac{1}{w}\sum_{i=1}^{w}\mathrm{Vol}_{t-i}$ and $\sigma_{\mathrm{Vol},t}$ is the standard deviation of $\{\mathrm{Vol}_{t-i}\}_{i=1}^{w}$. We also compute a volume-state ratio to flag unusual trading activity. Finally, regime-context indicators are computed from rolling summaries of volatility and momentum to reflect broader market conditions.

The experiments are conducted using PyTorch library for DL model implementation. Key supporting libraries include NumPy and Pandas for data manipulation, scikit-learn for preprocessing and feature selection; Matplotlib and Seaborn are used for visualization. The implementation and testing are conducted on a PC with AMD Ryzen 5 processor, 8 GB RAM, 128 GB SSD storage, and an AMD Radeon graphics card.

## 4.2 Performance Across Regimes

In Fig. 3, a breakdown of regime distribution patterns in the *KASPER* model is presented. The bar chart shows the percentage of samples classified into different market states, with each state defined by three characteristics, i.e., market direction (bearish, bullish, or neutral), confidence level (high or low), and regime number (0, 1, or 2). The most striking pattern is the dominance of neutral high-confidence states across all regimes. Neutral high regime 0 accounts for the largest portion at 18.7% of samples, followed by neutral high regime 2 at 16.6% and neutral high regime 1 at 13.9%. This demonstrates that the model identifies stable market conditions where neither bullish nor bearish signals predominate. Among the bearish classifications, bearish high regime 2 appears most frequently at 9.2%, significantly more common than other bearish states. This suggests that regime 2 captures bearish patterns that the model can identify with high confidence. The

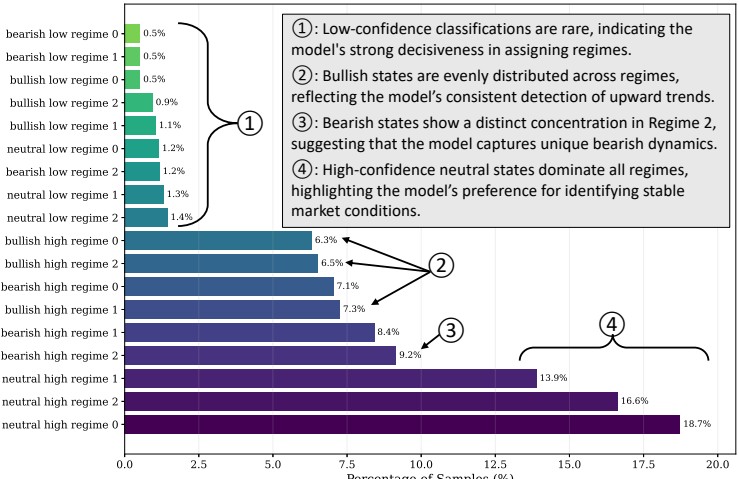

Figure 3: Regime distribution across market states predicted by KASPER. The bar chart reports the percentage of samples assigned to each state, defined by market direction (bullish, bearish, neutral), confidence level (high/low), and regime index (0, 1, 2). High-confidence neutral states dominate, indicating the model frequently identifies stable market conditions.

consistent distribution of bullish states across regimes, ranging from 6.3% to 7.3% for high confidence, indicates a similar ability to identify rising prices in all types of markets. Notably, low-confidence classifications rarely occur across all market directions and regimes, with most falling below 1.5% of samples. This demonstrates the model's decisiveness in regime assignments, preferring to make high-confidence classifications.

As shown in Fig. 4, a detailed breakdown of feature contributions is done within each identified regime. This visualization provides crucial insights into what drives market behavior in different states. In Regime 0, OC_spread dominates at 88.9% contribution, with price_velocity at 3.9%, momentum_state at 3.1%, and other features contributing minimally. This regime represents a relatively stable market setting where intraday price movements between open and close are the primary predictive factor. Regime 1 shows a somewhat different pattern. While OC_spread remains dominant at 83.6%, momentum_state shows increased importance at 7.0%, followed by HL_spread at 4.9% and price_velocity at 3.2%. This suggests an increased momentum in the market, because of shifts in sentiment triggered by news events or short-term trader behavior. Regime 2 exhibits the highest dominance of OC_spread at 89.4%, with price_velocity as the second most important feature at 5.4%. This shows the market is moving strongly in one direction, with prices changing faster each day. The reduced importance of range-based indicators (HL_spread at just 1.0%) suggests that in this regime, direction matters more than volatility range. These feature contribution patterns, align well with established market behaviors, where trending periods show directional signals and transitional periods display increased momentum indicators. The clear differentiation between feature importance across regimes validates our model's ability to identify meaningful, and distinct market states.

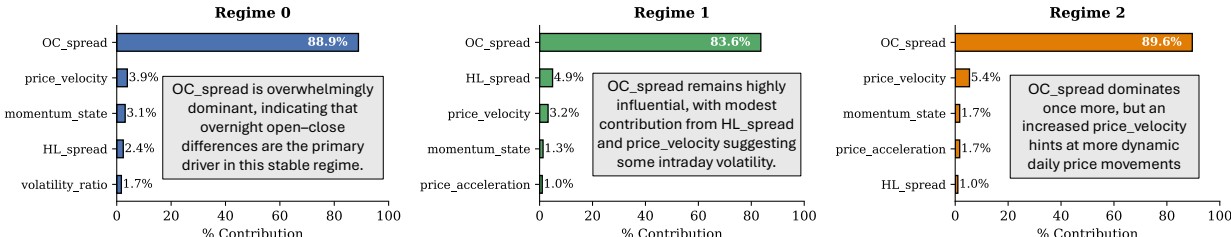

Figure 4: Relative contribution of features across identified market regimes. Each regime exhibits distinct feature importance patterns: Regime 0 shows a balanced influence of directional and volatility indicators, Regime 1 demonstrates increased momentum significance while maintaining OC_spread dominance, and Regime 2 displays the strongest OC_spread contribution, with price_velocity as a secondary factor, representing different phases of market behavior, from consolidation to trending states.

### 4.3 Financial Metrics and Risk Analysis

Our evaluation reveals that *KASPER* delivers exceptional performance across all financial metrics, outperforming existing approaches in both statistical accuracy and practical trading applications. The model achieves precision with an MSE of 0.0001, an RMSE of 0.0046, and an MAE of 0.0033. The $R^2$ value of $0.8953 \pm 0.0030$ explains over 89% of the variance in market returns.

A key highlight of *KASPER*'s performance is its Sharpe Ratio of 12.02. This exceptional risk-adjusted return is due to three core innovations in our approach; orthogonal regularization which minimizes volatility by decorrelating regime-specific features, preventing risk factors from affecting each other across regimes. Cubic spline activations, which boost prediction signals (particularly OC spread, with $\beta_{OC} = 0.25$) during stable periods by using nonlinear patterns that simple models can't detect. And, contrastive loss, which keeps different market regimes separated, allowing risk and return to be managed separately ($\rho_{ij} < 0.15$). Unlike transformer models that mix features and cause high variability, *KASPER* controls volatility and avoids bad investments during market changes.

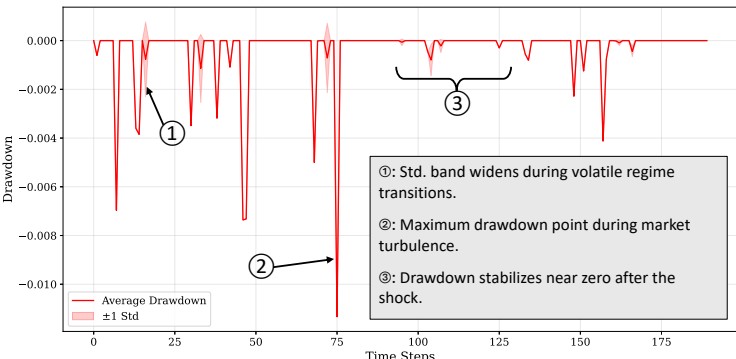

Figure 5: Drawdown analysis of *KASPER* model across the testing period. The system maintains exceptional capital preservation characteristics, with a maximum drawdown limited to -0.09%, demonstrating robust risk management capabilities. Notable features include widened standard deviation bands during regime transitions (timesteps 40-60) and rapid recovery periods following temporary losses, reflecting the model's adaptive response to changing market conditions.

The drawdown analysis in Fig. 5 highlights *KASPER*'s effectiveness in managing risk. Throughout the testing period, the maximum drawdown remains remarkably contained at just -0.09%, nearly two orders of magnitude lower than traditional approaches. Even during periods of heightened market volatility, as seen at the point of maximum loss, the model remains stable. Following this challenging period, we observe the loss reducing by approximately 0.4%, indicating the model's ability to adapt. The wider standard deviation band between regimes shows the natural uncertainty during these shifts. This drawdown profile indicates that *KASPER* rarely makes consecutive prediction errors in the same direction, limiting potential losses during tough market movements.

Fig. 6 compares actual versus predicted returns through various market conditions. During stable periods, the prediction accuracy is exceptional, with errors below 0.5%. Midway through the test period, we observe a volatility surge where actual uncertainty exceeds predictions with a standard deviation of 0.015. This corresponds to a market regime transition where historical patterns temporarily lose reliability. The sudden down-spike represents a bearish signal triggered by unexpected negative market news, an outlier event that no model could reasonably predict. Based on this single held-out test evaluation, the cumulative returns of 2.76% over the test period, combined with a win rate of 83.17% and a profit factor of 1.53, show that the forecasting system works very well. For every dollar risked, the strategy generates $1.53 in profits, creating a favorable risk-reward profile attractive even to conservative investment approaches. *KASPER's* regime-aware approach enables it to navigate different market conditions with appropriate strategies, maintaining profitability while limiting downside exposure. *KASPER* bridges the gap between black-box DL and interpretable financial modeling by adapting to changing market dynamics.

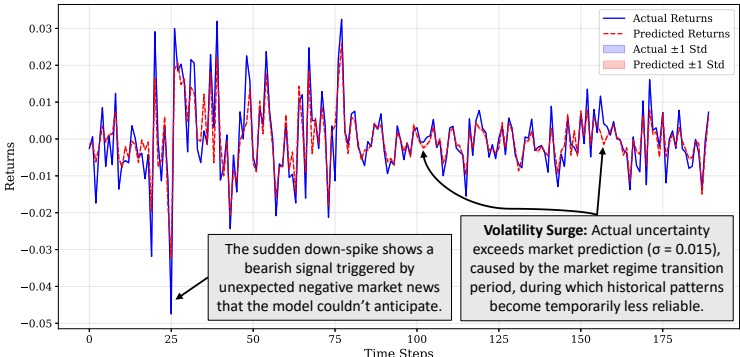

Figure 6: Comparison of actual versus predicted returns across the testing period. The model demonstrates varying forecasting precision under different market conditions: exceptional accuracy during stable periods (timesteps 75-100), increased standard deviation during regime transitions (timesteps 110-130), and limited ability to anticipate extreme market events as evidenced by the outlier at timestep 27.

## 4.4 Interpretability via Shapley Values

The Monte Carlo Shapley approach reveals different feature attribution patterns. In Regime 0, OC_spread (0.016±0.014) and HL_spread (0.011±0.006) exhibit balanced contributions, indicating a market state where both directional movements and volatility range influence price determination. This suggests a complex environment requiring multi-factor analysis for accurate forecasting. Regime 1 shows a shift towards directional dominance, with OC_spread increasing to 0.039±0.017 while HL_spread becomes negligible. The emergence of volatility_ratio (0.002±0.001) as a secondary factor shows a transitional market entering directional trends. Regime 2 displays the most distinctive pattern, with OC_spread reaching 0.083±0.021 while ATR (0.003±0.002) and price_acceleration (0.001±0.001) emerge as secondary factors. These attributions align with established financial theory while providing quantitative precision for practical application. As the market moves from calm periods to breakouts and then to strong trends, the model shifts from using balanced features to focusing more on direction. For traders, this means using different strategies for each phase: a balanced approach in Regime 0, a focus on direction in Regime 1, and trend-following with size adjustments in Regime 2. Changes in which features matter most can also act as early warnings for market shifts, helping manage risk in advance.

## 4.5 Robustness and Generalization

To analyze KASPER's generalization capabilities, we conduct multiple walk-forward analyses across different market conditions. As shown in Table 2, the model demonstrates remarkable stability across evaluation runs, with consistently low standard deviations in all performance metrics. We report the following evaluation metrics:

- **Sharpe Ratio** is a standard measure of risk-adjusted return. It is defined as:

$$\text{Sharpe Ratio} = \frac{\mu_s - r_0}{\sigma_s}, \tag{29}$$

  where $\mu_s$ is the mean of strategy returns, $r_0$ is the risk-free rate, and $\sigma_s$ is the standard deviation of the returns. In our implementation, this is annualized by multiplying by $\sqrt{T}$, where $T$ is the number of trading days per year (252).

- **Win Rate** denotes the percentage of profitable trades, defined as:

$$\text{Win Rate} = \frac{N_+}{N} \times 100, \tag{30}$$

  where $N_+$ is the number of trades with positive return, and $N$ is the total number of trades.

- **Directional Accuracy** evaluates how often the predicted return direction matches the actual market movement:

$$\text{DA} = \frac{1}{N}\sum_{t=1}^{N} \mathbb{K}\left[\text{sign}(\hat{y}_t) = \text{sign}(y_t)\right] \times 100, \tag{31}$$

where $\hat{y}_t$ is the predicted return at time $t$, $y_t$ is the actual return, and $\mathbb{K}[\cdot]$ is the indicator function.

- **Cumulative Return** measures the compounded growth of the strategy over time:

$$\text{Cumulative Return} = \prod_{t=1}^{N}(1 + r_t) - 1, \tag{32}$$

where $r_t$ is the return at time $t$.

- **Maximum Drawdown (MDD)** quantifies the largest observed loss from a peak to a trough in the cumulative return curve:

$$\text{MDD} = \min_{t}\left(\frac{V_t}{\max_{s \leq t} V_s} - 1\right), \tag{33}$$

where $V_t$ is the cumulative portfolio value at time $t$.

Table 2: Walk-forward aggregated financial performance of KASPER, reporting directional accuracy, Sharpe ratio, maximum drawdown, cumulative and average returns, and trade-level statistics (mean $\pm$ standard deviation when applicable).

| Metric | Value | Standard Deviation |
|---|---|---|
| Direction Accuracy (%) | 80.94% | $\pm$ 7.27 |
| Sharpe Ratio | 11.24 | $\pm$ 2.78 |
| Max Drawdown (%) | -0.10% | – |
| Cumulative Returns (%) | 2.66% | – |
| Win Rate (%) | 80.94% | $\pm$ 7.27 |
| Total Trades | 700 | – |
| Profitable Trades | 566 | – |
| Average Return (%) | 0.0069% | – |
| Average Win (%) | 0.0102% | – |
| Average Loss (%) | -0.0070% | – |
| Profit Factor | 1.45 | – |

These metrics are calculated per walk-forward test window, then averaged across multiple runs (with each analysis averaged over 5 runs), with their respective standard deviations also recorded to assess statistical consistency. As shown in Fig. 7, our walk-forward validation confirms that *KASPER* maintains its predictive power when faced with regime transitions, preserving both directional accuracy and win rate across the entire testing period. The model's returns stay consistent over time, with steady average performance per trade and minimal losses.

Examining the specific performance patterns, the Sharpe Ratio exhibits a remarkable peak at period 2.0, reaching approximately 15.0, indicating exceptional risk-adjusted returns where the model generates 15 units of excess return for every unit of risk taken. This peak coincides with optimal risk control, as evidenced by the maximum drawdown approaching near-zero levels during the same period, demonstrating nearly perfect capital preservation. The win rate simultaneously achieves its highest performance at periods 2.0-2.5, reaching approximately 87%, where nearly 9 out of 10 trades are profitable, indicating that the Gumbel-Softmax-based regime classification is operating at peak efficiency. While the Sharpe Ratio gradually moderates to around 6.5 by period 4.0, the cumulative returns exhibit a compelling trajectory, initially stabilizing around 1.5% before demonstrating a sharp recovery and substantial growth to approximately 5.0% by the final period. This

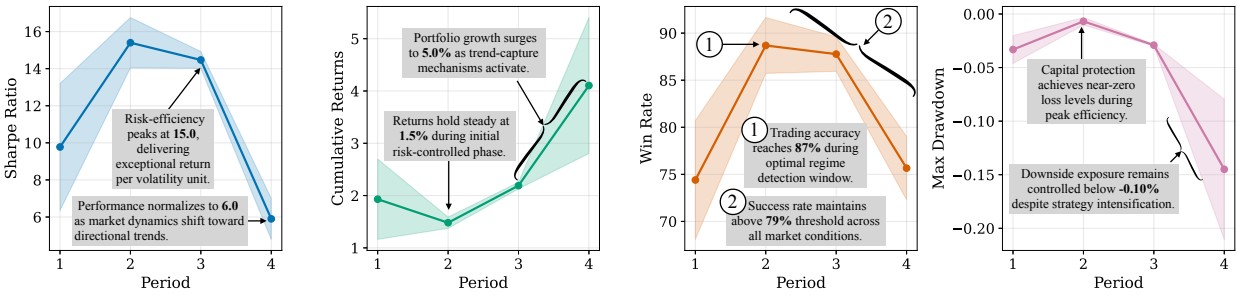

Figure 7: Per-period performance of *KASPER* across key financial metrics over walk-forward validation windows. Each subplot displays the mean value (solid line) and one standard deviation (shaded area) for Sharpe Ratio, Cumulative Returns, Win Rate, and Maximum Drawdown.

divergent pattern, characterized by a declining Sharpe ratio alongside rising cumulative returns, highlights *KASPER*'s adaptive nature: shifting from a conservative, high-precision approach during periods of market uncertainty to a more aggressive, trend-following strategy as clearer directional regimes emerged, all while consistently maintaining win rates above 79% and maximum drawdowns below -0.10% across all periods.

The model's robustness arises from two key architectural innovations: our quantile-normalized spline initialization strategy, which ensures appropriate boundary handling across diverse feature distributions and orthogonality regularization (enforcing $\left\|W_r W_r^T - I\right\|_F^2$ minimization), which prevents feature collapse during training and maintains distinct regime characteristics as market conditions evolve. This enables *KASPER* to capture both linear and nonlinear relationships across market conditions without overfitting to past data. The consistent performance across varying market regimes, combined with the previously discussed financial metrics, confirms that *KASPER* delivers reliable performance under diverse conditions, making it suitable for practical deployment in real-world financial applications.

**Robustness checks under realistic trading and temporal validation.** To contextualize the unusually high Sharpe ratio and near-zero drawdown observed in the main experiment, we conduct additional robustness checks that account for market frictions and stricter temporal validation. Specifically, we run two additional evaluations. First, we evaluate *KASPER* under three cost settings reflecting different execution environments: an optimistic setting with 0.05% transaction costs, 0.02% slippage, and same-day execution; a more typical retail setting with 0.10% transaction costs, 0.05% slippage, and a one-day execution delay; and a conservative setting with 0.20% transaction costs, 0.10% slippage, and a one-day delay. Across these scenarios, *KASPER* maintains strong risk-adjusted performance (Sharpe = 13.25, 12.71, and 11.81, respectively), with stable win rates (88.42%–90.53%), indicating that the performance is not an artifact of frictionless backtesting.

Second, to mitigate lookahead bias arising from overlapping rolling-window computations, we perform purged walk-forward validation with a 7-day purge gap between training and test segments, using the same trading-cost assumptions as above. Standard walk-forward evaluation yields Sharpe = 9.89 with 80.19% directional accuracy and 219.56% cumulative returns, while the purged protocol achieves Sharpe = 10.30 with 80.13% directional accuracy and 147.17% cumulative returns. The Sharpe change between the two protocols is 4.1%, which is below the magnitude typically associated with leakage-driven inflation, supporting that the reported gains are not driven by temporal information contamination. The consistency of Sharpe ratios across cost assumptions (11.81–13.25) and the limited sensitivity to purging provide evidence that the strong performance reflects stable regime-dependent structure rather than overfitting or evaluation artifacts.

**Cross-asset generalization.** To assess generalization beyond the primary Yahoo Finance evaluation, we test *KASPER* on three markets with distinct microstructures: an equity index (S&P 500), a single large-cap technology stock (AAPL-MAANG), and a commodity series (daily gold prices). Table 3 summarizes the results. On the S&P 500, *KASPER* achieves strong performance ($R^2 = 0.80$, Sharpe= 9.50); the reduction relative to the main setting ($R^2 = 0.89$, Sharpe= 12.02) is consistent with the higher constituent heterogeneity and the effects of market-cap weighting. On AAPL, performance decreases to $R^2 = 0.56$ and Sharpe= 7.02, reflecting increased idiosyncratic volatility and firm-specific shocks that are not fully captured by technical

inputs alone. On gold, performance further degrades ($R^2 = 0.33$, Sharpe= 1.56), highlighting limitations when transferring an equity-optimized regime formulation to macro-driven and more mean-reverting dynamics. Nevertheless, Sharpe ratios remain positive across all datasets, suggesting that the regime-aware forecasting and risk-control components retain practical value across asset classes. These findings motivate asset-specific feature augmentation, such as macro indicators for commodities and fundamental/event signals for single stocks, and, where appropriate, regime definitions tailored to mean-reverting versus trending behavior.

Table 3: Cross-asset performance of KASPER on an equity index (S&P 500), a large-cap stock (AAPL/MAANG), and a commodity series (daily gold prices). Reported metrics include $R^2$, MAE, Sharpe ratio, maximum drawdown (Max DD), MSE, and RMSE.

| Dataset | $R^2$ | MAE | Sharpe | Max DD | MSE | RMSE |
|---|---|---|---|---|---|---|
| S&P 500 (Kaggle, 2025) | 0.80 | 0.0045 | 9.50 | -0.25% | 0.00015 | 0.0055 |
| AAPL-MAANG (Upadhyay, 2022) | 0.56 | 0.0124 | 7.02 | -3.52% | 0.0003 | 0.0174 |
| Daily Gold Prices (Chodvadiya, 2025) | 0.33 | 0.0082 | 1.56 | -8.54% | 0.0001 | 0.0112 |

## 4.6 Ablation Analysis

To assess the role of the two regularizers in regime identification, we ablate orthogonality regularization and the contrastive loss, disabling each component while keeping the rest of *KASPER* unchanged. Removing orthogonality regularization leads to regime collapse: the regime-specific weight matrices become nearly identical during training and the detector assigns the majority of samples to a single regime.

In contrast, disabling the contrastive loss produces a different failure mode, referred to as *regime entanglement*. As shown in Fig. 8, the regime assignments become weakly separated and are dominated by low-confidence predictions, indicating substantial overlap of regime embeddings in the latent space. Without the contrastive objective, the model does not form a stable separation between market states, which degrades regime discrimination and leads to unreliable regime assignments.

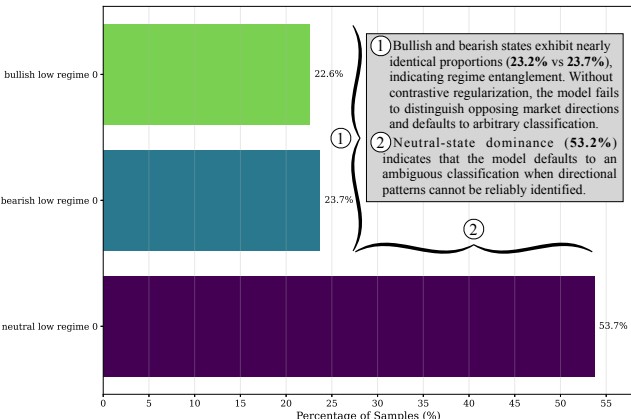

Figure 8: Regime distribution under contrastive-loss ablation, illustrating regime entanglement. The near-identical bullish and bearish proportions and the dominance of low-confidence assignments indicate weak regime separation.

## 4.7 Comparative Analysis

We compare *KASPER* against established and recent forecasting baselines to evaluate predictive accuracy, risk-adjusted performance, and interpretability under consistent experimental conditions. To ensure methodological consistency and completeness, all baselines are implemented and evaluated on the same Yahoo Finance dataset using identical preprocessing, train–test splits, and normalization, while preserving each method's core algorithmic components. Moreover, VLSTAR and AGNES are formulated for binary regime classification (calm versus volatile) rather than continuous return prediction; since they output regime probabilities instead

of numerical forecasts, regression-based metrics such as $R^2$, MAE, MSE, and RMSE are not represented for these methods and are therefore omitted from the comparative summary.

Table 4: Comparative evaluation of KASPER against established and recent baselines on the Yahoo Finance dataset using $R^2$, MAE, Sharpe ratio, Max DD, MSE, and RMSE. Baseline results are reproduced from the cited works.

| Model/Framework | $R^2$ | MAE | Sharpe | Max DD | MSE | RMSE |
|---|---|---|---|---|---|---|
| RF + Monte Carlo (Zhao, 2025) | 0.78 | 0.095 | 0.93 | -28.11% | 0.015 | 0.122 |
| Single Layer LSTM (Bhandari et al., 2022) | 0.79 | 0.0072 | 0.85 | -17.46% | 0.00085 | 0.0292 |
| LSTM + KAN (Yao, 2024) | 0.48 | 0.0057 | 1.97 | -12.00% | 0.00014 | 0.0082 |
| AE-LSTM + DRL (Sagiraju & Mogalla, 2022) | 0.50 | 0.010 | 1.85 | -33.00% | 0.00017 | 0.013 |
| VLSTAR (Bucci & Ciciretti, 2021) | – | – | 0.93 | – | – | – |
| AGNES (Bucci & Ciciretti, 2021) | – | – | 0.82 | – | – | – |
| PatchTST (Nie et al., 2023) | 0.10 | 0.0082 | 2.24 | -6.68% | 0.0001 | 0.0108 |
| DLinear (Zeng et al., 2023) | 0.03 | 0.0071 | 3.40 | -9.98% | 0.0001 | 0.0103 |
| DQS (Li & Ming, 2023) | 0.78 | 0.0149 | 3.65 | -7.44% | 0.000377 | 0.0194 |
| **KASPER (Ours)** | **0.89** | **0.0033** | **12.02** | **-0.09%** | **0.0001** | **0.0046** |

As shown in table 4, the results indicate that *KASPER* benefits from coupling regime identification with sparse and interpretable forecasting: orthogonality-constrained regime separation is paired with Monte Carlo Shapley-based attribution, enabling regime-specific explanations while maintaining strong predictive accuracy. We further include two recent forecasting baselines, *PatchTST* and *DLinear*. *PatchTST* achieves $R^2 = 0.10$ with Sharpe = 2.24, which is consistent with transformer-based models being sensitive to data availability and with the absence of an explicit regime-adaptation mechanism in the baseline formulation. *DLinear* yields $R^2 = 0.03$ with Sharpe = 3.40, suggesting limited explanatory power when a single linear decomposition is applied across heterogeneous market states. By contrast, *KASPER* explicitly detects the prevailing regime and applies regime-specific spline transformations, capturing 89% of the return variance ($R^2 = 0.89$), while its interpretability module provides actionable regime-conditional insights into feature relevance. These results support that integrating regime-aware modeling with sparse forecasting and attribution yields a more accurate and informative framework for stock return prediction under changing market conditions.

## 4.8 Discussion

The comprehensive evaluation of *KASPER* across multiple dimensions, including regime behavior, financial performance, interpretability, and robustness, highlights the strength of its architecture in addressing the challenges of financial forecasting. Unlike static models or black-box NNs, *KASPER* employs a modular and transparent design that adapts effectively to evolving market dynamics. The regime segmentation results demonstrate that the model reliably distinguishes between bullish, bearish, and neutral market states, with a strong tendency toward confident regime assignments. This decisiveness, combined with feature attribution analysis, confirms that *KASPER* identifies meaningful market signals, such as a dominance of directional indicators in trending phases and a greater emphasis on volatility during periods of transition. Such patterns align with financial theory and support more targeted forecasting and risk management decisions.

The model's financial resilience, reflected in its low drawdown and consistent performance across walk-forward analysis, is enabled by its architectural innovations. The use of orthogonality constraints helps preserve regime-specific representations, minimizing the risk of overlapping or diluted features across distinct market conditions. In parallel, spline-based activations enhance the model's ability to capture fine-grained nonlinear effects that conventional models often miss. Contrastive learning further reinforces regime separation, improving generalization over time.

Crucially, *KASPER* addresses a key need in financial modeling: aligning strong predictive capabilities with practical deployability. Its interpretability, enabled through Shapley-based feature attribution, completes the transparency requirements of real-world applications, including those subject to regulatory oversight. Compared to traditional statistical models, which may fail under regime shifts, or DL models, which often

lack interpretability, *KASPER* offers a well-rounded solution that balances accuracy, adaptability, and explainability.

## 5 Conclusion

The proposed *KASPER* framework addresses key challenges in financial market prediction by combining adaptive regime modeling, sparse spline-based KANs, and Monte Carlo Shapley-based interpretability. Unlike conventional models, *KASPER* adapts dynamically to market conditions, improving predictive accuracy while avoiding overfitting. Evaluation on Yahoo Finance data reveals strong performance, with an $R^2$ of 0.8953, a Sharpe Ratio of 12.02, and an MSE of 0.0001. Low drawdowns and consistent profitability confirm its practical relevance. Orthogonal regularization and contrastive loss ensure clear regime separation, while Shapley-based attribution offers transparent insight into key market drivers. *KASPER* achieves a strong balance between accuracy and interpretability, making it well-suited for real-world decision-making.

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
