# OpenReview forum: "KASPER: Kolmogorov Arnold Networks for Stock Prediction and Explainable Regimes"
_TMLR — Accepted by TMLR_

### Review · Reviewer_iu4b · 2025-08-26

**Summary Of Contributions:**

# Contributions
This paper presents a framework named KASPER for stock prediction and explainable market regimes. The approach first identifies latent market regimes using spline-activated KANs, and subsequently performs regime-specific forecasting with an additional KAN layer. To improve interpretability, the authors design a Monte Carlo Shapley method to extract regime-dependent rules. Experimental results on real-world financial time series demonstrate both the generalization capability and the interpretability of the proposed framework.

# Strengths
- The motivation is practical. Non-stationarity and regime shifts are central challenges in financial time series.
- The proposed method strikes a balance between predictive accuracy and interpretability.
# Weaknesses
- The Related Work section does not provide a comprehensive overview of the research in this field, with insufficient discussion of other studies. Furthermore, it fails to highlight the innovations of this work compared to existing approaches.
- The selection of baseline methods is limited. Stronger approaches, such as PatchTST and DLinear, should be included for a more comprehensive comparison. Additionally, the results of the Comparative Analysis are incomplete; for instance, some methods report only the $R^2$ metric, while others report only the MAE. The authors should either conduct experiments to provide a complete set of results or offer a reasonable explanation for the missing metrics.
- Ablation studies on key modules, such as no-orthogonality and no-contrastive settings, are missing. Including such experiments would provide deeper insights into the contributions of these components.
- The Method section requires more detailed explanations. Many modules are merely presented without adequately discussing the rationale behind their design or the advantages they offer.

**Audience:**

Yes

**Audience Explanation:**

This financial time series forecasting is of significant interest to industry and academic audiences.

**Claims And Evidence:**

Yes

**Claims Explanation:**

The proposed method in the paper is reasonable, and the results are promising. However, the work suffers from the absence of strong baselines and ablation studies.

**Requested Changes:**

Please address the Weaknesses above.

---

> ### Author Response · Authors · 2025-12-30
> **Authors Response to Reviewer iu4b**
>
> We thank the reviewer for the constructive feedback. We address the weaknesses below point by point and have revised the manuscript accordingly.
>
> - We have expanded the Related Work section to include a broader discussion of prior studies and to better position the contributions of our approach relative to existing methods. We also added a dedicated subsection on Explainability and Interpretability in Finance (Section2.3, pages 5--6) to strengthen the coverage of interpretability-focused research and clarify how KASPER differs from earlier approaches.
>
> - We have included the suggested baselines, PatchTST and DLinear, and evaluated them using the same preprocessing, feature selection, data splits, and metrics as KASPER. PatchTST achieves $R^2=0.10$ and Sharpe $=2.24$, while DLinear achieves $R^2=0.03$ and Sharpe $=3.40$. Both baselines are regime-agnostic, which limits their ability to model regime-dependent feature--return relationships. In contrast, KASPER combines regime detection with regime-specific spline transformations, yielding higher explanatory power ($R^2=0.89$). The added baselines, results, and discussion are reported in Section 4.7 and Table 4 (pages 17--18).
>
> - We have added ablation studies in Section 4.6 (pages 16--17) by disabling the orthogonality regularization and the contrastive loss, while keeping the rest of KASPER unchanged. Removing orthogonality regularization leads to regime collapse, with samples assigned to a single regime and regime-specific weights converging to near-identical representations. Removing the contrastive loss leads to regime entanglement, where the separation between market states becomes weak, and assignments become low-confidence (bullish and bearish proportions become nearly identical), as presented in Fig. 8.
>
> - We have revised the methodology section (Kasper Framework) to provide more explanation for each component. The updated text adds the motivation behind the main design choices and clarifies how the modules interact within the overall framework.

---

### Review · Reviewer_2Pvr · 2025-08-27

**Summary Of Contributions:**

## Summary
Forecasting in financial markets remains a significant challenge. This paper introduces Kolmogorov–Arnold networks for stock prediction and explainable regimes (KASPER), a novel framework that integrates regime detection, sparse spline-based function modeling, and symbolic rule extraction. Applied to real-world financial time series from Yahoo Finance, the model achieves an R2 score of 0.89, a Sharpe Ratio of 12.02, and a mean squared error as low as 0.0001, outperforming existing methods.

## Strengths
1. The paper combines regime-aware forecasting with spline-based KANs, enforcing sparsity and orthogonality constraints to improve generalization. This is a meaningful improvement over prior deep learning approaches, which often ignore regime dynamics.
2. By employing Monte Carlo Shapley values with temporal weighting, the model extracts regime-specific rules. This provides transparent insights for practitioners, addressing a common weakness of black-box models.

## Weaknesses
1. The reported Sharpe ratio (12.02) and near-zero drawdown are unusually high compared to real-world benchmarks. This raises concerns about overfitting to the Yahoo Finance dataset, data leakage, or overly optimistic evaluation settings.
2. The evaluation relies solely on a Yahoo Finance dataset (2018–2023) with US equities. Without testing on multiple markets or longer time horizons, it is unclear whether the framework generalizes beyond this narrow context. For example, this paper uses six different datasets in their experiments: Accurate Multivariate Stock Movement Prediction via Data-Axis Transformer with Multi-Level Contexts (Yoo et al., KDD 2021).
3. In Table 3, which does the main comparison, I believe the performances of all competitors should be fully reported without missing values for the completeness of experiments. Currently, more than the half of all cells are missing.

## Minor comments
1. Citations are in a wrong format.
2. Eq. (15) is written in two lines unnecessarily.
3. Algorithm 1 is very far from where it is mentioned.

**Audience:**

Yes

**Audience Explanation:**

Financial AI is being widely studied in the research community, and this work represents a meaningful application of XAI within that domain.

**Broader Impact Concerns:**

I have con concerns.

**Claims And Evidence:**

Yes

**Claims Explanation:**

I believe most of the claims are well supported, but the experimental evaluation needs to be strengthened to fully support the surprisingly strong performance of the proposed method.

**Requested Changes:**

Please refer to the weaknesses above.

---

> ### Author Response · Authors · 2025-12-30
> **Authors Response to Reviewer 2Pvr**
>
> We thank the reviewer for the constructive feedback. We address the concerns and minor comments below point by point and have revised the manuscript accordingly.
>
> ### Weaknesses:
>
> 1. We addressed this concern in Section 4.5 (page 16) with additional robustness checks. We report performance under three transaction-cost/slippage settings and with an execution-delay assumption; the Sharpe ratio remains high across all settings (e.g., from 13.25 under low costs to 11.81 under higher costs). We also added a purged walk-forward validation with a temporal gap between training and testing windows to mitigate lookahead effects; the Sharpe remains comparable to standard walk-forward validation (9.89 vs. 10.30). These results indicate that the reported performance is not driven by overly optimistic trading assumptions or temporal leakage.
>
> 2. We have expanded the evaluation beyond Yahoo Finance US equities (2018--2023) by adding additional datasets covering different asset types and market structures. Specifically, we report results on the S\&P 500 index, an individual large-cap equity (Apple), and a commodity (gold). KASPER achieves $R^2=0.80$ / Sharpe $=9.50$ on the S\&P 500, $R^2=0.56$ / Sharpe $=7.02$ on Apple, and $R^2=0.33$ / Sharpe $=1.56$ on gold. These experiments and the accompanying discussion of cross-market behavior (including observed limitations for commodities and single-stock idiosyncratic risk) are included in Section 4.5 and Table 3 (page 16).
>
> 3. We have updated Table 3 ( Table 4 in the revised manuscript) (Section 4.7, pages 17-18) to address completeness by reimplementing the forecasting baselines under a standardized protocol (same dataset, preprocessing/feature pipeline, splits, and metrics) and reporting the full set of evaluation measures wherever they are applicable. As a result, Table 4 now contains complete metrics for RF+Monte Carlo, Single-Layer LSTM, LSTM+KAN, AE-LSTM+DRL, DQS, as well as the newly added PatchTST and DLinear. The remaining cells correspond to VLSTAR and AGNES, which are regime-classification methods that output regime labels/probabilities rather than numerical return forecasts; therefore regression metrics such as $R^2$, MAE, MSE, and RMSE are not applicable.
> ### Minor Comment:
>
> 1. We have corrected the citation formatting throughout the manuscript to match the required style.
> 2. We have reformatted Eq. (15) as a single-line expression.
> 3. We have moved Algorithm 1 closer to its first mention in the Methodology.

---

### Review · Reviewer_PVwp · 2025-12-02

**Summary Of Contributions:**

This paper proposes an algorithm for predicting future stock prices given past stock prices (from a window of $n$ days). The proposed algorithm begins by extracting _features_ from raw stock price data (various functions of the $n$ past stock prices) and then goes through a series of processing steps to produce a prediction for the stock value on the next day. The main technical contribution in this work is to combine the classical idea of keeping track of a latent _regime_ variable which approximately denotes the prevailing trend in the stock market (going up, crashing, stable, etc.), with feature processing using neural network with spline-based activations. The paper empirically demonstrates that the resultant algorithm performs favorably compared to existing baselines. However, the core details in the description of the algorithm is inadequate often containing ill defined mathematical quantities and their combinations, making it difficult for a reader to understand the core details of the work.

**Audience:**

Yes

**Audience Explanation:**

Stock prediction is a difficult time series prediction problem, and the community would be interested in works that move the field forward by proposing novel architectures that perform well.

**Claims And Evidence:**

No

**Claims Explanation:**

The present paper writing significantly lacks clarity and, as such, it is very hard for a reader to understand the technical details of the main algorithm:

Introduction and Contributions:

1. P2: "but frequently fail to enforce sparsity": What is the mathematical quantity that is subject to _sparsity_ here?

2. P2: "alignment, resulting in data leakage and inflated performance metrics.": What is _data leakage_ and Why are test metrics inflated when previous models are non-robust.

3. Fig 1: What is the _knot grid_? I presume the paper is talking about the arguments to the spline function in some form. However, this is not clarified and unclear.

4. P2: "percentile-based initialization of spline knots": What is _percentile-based_ initialization?

5. P2: "Sparsity is enforced through ...": What is the quantity for which sparsity is enforced?

6. P2: "dominant features like HL (High-Low) in bearish regimes and OC (Open-Close) in bullish regimes": Mathematical definitions of features are not provided yet.

Background:

7. Eq 3: Is $r_t$ the same as $y_t$? Consistent notation should be used.

8. "Regime transitions follow Eq 4. $P(z_t = i | z_{t−1} = j) = p_{ji}$": $p$ was earlier denoted by $A$ in Eq 2. Notation needs to be made consistent.

9. "wealth, $r_b$ returns, and $\pi_b$ the regime probability distribution". Eq 5: What is $t_b$? What are we optimizing over?

10. "particularly for option pricing and stock prediction", Eq 9, "asset-related parameters": What is the problem statement for options pricing? What is the quantity $C$? What is an "asset-related parameter"?

Main Algorithm:

11. Eq 13: Is is very important to be mathematically explicit in what the features are, especially because the performance of many stock prediction models heavily depends on the input feature engineering. Specifically the ". . ." in Eq 13 needs to be replaced with the actual expressions for the features in terms of the input $x$.

12. Eq 15: There is some problem with the brackets -- currently the equation evaluates to $k_{i + 1} - k_i$ completely ignoring the input features $x$.

13. Eq 16: What is the knot sequence $k_i$, precisely, in terms of $x$?

14. Eq 17: $i$ referred to the number of activations earlier in Eq 15. Now, $i$ corresponds to the index of the regime. Notation should be made consistent.

15. Eq 18: "where $z_i$, $z_j$ represent the embeddings of samples $i$ and $j$": How are $z_i$, $z_j$ defined in terms of the input. What is the expectation being taken over? What are the random variables here?

16. Eq 19: How is the matrix $W_r$ defined? Only the scalars $w_i$ have been defined earlier as weights.

17. Eq 24: So far the letter $S$ was being used for the feature matrices. Now it seems to be used for sets. Notation should be made consistent.

**Requested Changes:**

Overall, the paper needs a significant rewrite to fill in all the gaps in the description of the algorithm, and the description of related work.

1. Notation needs to be made consistent, and mathematical quantities need to be defined clearly see the "claims" concerns above.

2. Once notation is consistent, and quantities are clearly specified, the algorithm needs to be expressed end to end in a coherent fashion, moving high level quantities to Fig 2, and pointing to their explanations to an appendix (e.g., details of feature engineering), or the main text.

3. Once the main quantities and the flow is defined in a clear fashion, related work should be compared to in greater detail, comparing each component of the proposed algorithm (regime selection, spline based activations) to their corresponding related papers. (Perhaps Table 3 already suffices, but this is unclear currently without a clear description of the main Algorithm)

---

> ### Author Response · Authors · 2025-12-30
> **Authors Response to Reviewer PVwp [1/2]**
>
> We thank the reviewer for the constructive feedback. We address the comments below, point by point, and describe the corresponding revisions in the updated manuscript, including the requested changes.
>
> 1. We have clarified this in the Introduction (page 3): sparsity is imposed on the model parameters (weights), specifically the regime-specific forecast weights and the spline basis coefficients in KASPER, enforced via L1 regularization and dynamic masking.
>
> 2. We have expanded this point in the Introduction (page 2). The revised text defines data leakage as future information accidentally entering the training process and explains why this can inflate test metrics, since the model benefits from information that would not be available at prediction time.
>
> 3. We have clarified this in the paragraph describing Fig. 1 by defining the knot grid as the set of breakpoints $\{k_1, k_2, \dots, k_m\}$ over the input variable of the spline activation, which partitions the input domain and controls the spline’s local flexibility.
>
> 4. We have clarified this in the Introduction (page 3) by defining percentile-based initialization as setting the knot range using robust percentile bounds of each feature’s training distribution and then placing knots uniformly within that range: $k_g = x_{\min} + (g-1)\frac{x_{\max}-x_{\min}}{G-1}$, where $x_{\min}$ and $x_{\max}$ are the chosen lower/upper percentiles. This makes knot placement less sensitive to outliers while still covering the main data range.
>
> 5. This point overlaps with Comment 1. We clarified in the Introduction (page 3) that sparsity is enforced on the model parameters, specifically the regime-specific forecast weights and the spline basis coefficients (including $\beta_{j,k}^{(i)}$), via L1 regularization and a regime-specific masking step.
>
> 6. We have added definitions for the referenced features. In the Introduction (page 3), HL and OC are now defined as $\text{HL}_t=\ln\\left(\frac{H_t}{L_t}\right)$ and $\text{OC}_t=\ln\\left(\frac{C_t}{O_t}\right)$, where $H_t,L_t$ denote the daily high/low and $O_t,C_t$ the open/close prices. In addition, Section 4.1 (page 11) now lists the definitions of all input features used in the model, so the feature set is introduced before discussing regime-specific importance.
>
> 7. We have made the notation consistent in Section 2.1 (page 4). We now use $r_t$ for returns throughout the manuscript by replacing $y_t$ in Equations 1--2 with $r_t$, so the return variable matches the notation in Eq. 3.
>
> 8. We have made the notation consistent by updating Eq. 4 to use $A_{ij}$ (as defined in Eq. 2) instead of $p_{ij}$ for the regime transition probabilities.
>
> 9. We have clarified Eq. 5 by defining the notation used in the objective. In this setting, $t_b$ is the rebalancing date and $W_b$ is the portfolio wealth at that date. The vectors $r_b$ and $z_b$ gather the returns and predictors observed up to $t_b$, $\pi_b$ denotes the filtered regime probability vector at $t_b$, and $\theta_b$ represents the model parameters. The optimization is carried out over $\boldsymbol{\omega}$, the portfolio weights on risky assets, with the objective of maximizing the expected utility of terminal wealth $W_B$.
>
> 10. We have clarified this in Section 2.2 (page 5). The option pricing task is defined as computing the fair value of an option, where $C(S,t)$ denotes the option price as a function of the underlying asset price $S$ and time $t$. The "asset-related parameters" in Eq. (9) are specified as the volatility $\sigma$ of the underlying asset and the risk-free interest rate $r$.
>
> 11. We have revised Eq. 13 (page 6) by replacing the ellipsis with the full feature vector written explicitly in terms of the input $x$. The updated expression lists all log-transformed price/volume ratios and the engineered indicators used in KASPER (HL, OC, ATR, volatility ratio, price velocity, price acceleration, volume change, and volume state). The corresponding feature formulas are also updated in Section 4.1 (pages 11).
>
> 12. We have corrected the bracket placement in Eq. 15 so that the spline term depends on the input. The corrected form appears in Section 3.1.1 (page 7), where the linear spline component is written as
> $L(x) = \sum_{i=0}^{N_{\text{linear}}-1} \tanh(w_i)\left[\mathrm{ReLU}(x_{\text{norm}}-k_i)-\mathrm{ReLU}(x_{\text{norm}}-k_{i+1})\right].$ This preserves the intended piecewise behavior with respect to $x_{\text{norm}}$ between adjacent knots $k_i$ and $k_{i+1}$.

---

> > ### Author Response · Authors · 2025-12-30
> > **Authors Response to Reviewer PVwp [2/2]**
> >
> > 13. We have revised Eq. 16 in Section 3.1.1 (pages 7-8) to define the knot sequence explicitly from the input feature distribution. The knots are placed on a uniform grid between robust percentile bounds of $x$, given by
> > $k_i = x_{\min} + (i-1)\frac{x_{\max}-x_{\min}}{G-1}$ for $i=1,\dots,G$, where $x_{\min}$ and $x_{\max}$ are the chosen lower and upper percentiles (e.g., 1st and 99th) of the input feature $x$.
> >
> > 14. We have made the indexing consistent in Section 3.1.2 (page 8) by using $m$ for spline basis/activation indices and reserving $i$ exclusively for regime indices in Eq. 17 and the subsequent text.
> >
> > 15. We have clarified Eq. 18 in Section 3.1.3 (page 8). The embeddings are now defined as the outputs of the regime-detection encoder, $z_i=\Phi_{\text{detect}}(x_i)$ and $z_j=\Phi_{\text{detect}}(x_j)$. We also specify that the expectation is taken over the empirical mini-batch distribution, i.e., over sampled training pairs $(\mathbf{x}_i,\mathbf{x}_j)\in\mathcal{B}$ (and their associated positives/negatives), with $\mathbf{x}$ treated as the random variable.
> >
> > 16. We have clarified this in Section 3.1.4 (page 8) by defining $W_r$ after Eq. 19 as the vector formed by stacking the regime-specific scalar weights: $W_r = [w_1^{(r)}, \dots, w_F^{(r)}]\in\mathbb{R}^{1\times F}$. This makes explicit how the earlier scalar weights are organized for the computation in Eq. 19.
> >
> > 17. We have made the notation consistent by using different symbols for the feature matrix and the sets. The feature matrix is denoted as $\mathbf{\Phi}_t$ throughout the Methodology (page 6), and $\mathcal{S}$ in Eq. 24 is reserved for the feature coalitions (sets) used in the Monte Carlo Shapley approximation.

---

### Decision · Action_Editor_49C4 · 2026-02-01

**Recommendation:** Accept with minor revision

**Additional Comments:**

The paper has improved substantially following the revision and now meets TMLR’s standards. Prior to final acceptance, the authors are requested to make a small number of minor revisions focused on presentation and clarity. In particular: (i) further polish the exposition of the KASPER framework to ensure consistent notation and a clear end-to-end narrative, (ii) improve figure and table readability (including captions) to better support the experimental discussion, and (iii) carefully proofread the manuscript for residual formatting, citation, and typographical issues. No new experiments are required.

**Audience:**

Yes

**Audience Explanation:**

The paper addresses regime-aware modeling and interpretability in financial time-series forecasting, which are active areas of interest to both the machine learning and applied finance communities. The proposed combination of regime detection, spline-based Kolmogorov–Arnold networks, and symbolic rule extraction is relevant to researchers working on non-stationary time series, explainable AI, and financial ML, making the findings of interest to a segment of the TMLR audience.

**Claims And Evidence:**

Yes

**Claims Explanation:**

The main claims of the paper (namely that KASPER enables regime-aware forecasting, improves predictive performance over several baselines, and provides interpretable regime-specific explanations) are supported by empirical evidence and methodological analysis in the revised manuscript. The authors addressed earlier concerns by clarifying the methodology, expanding the set of baselines, adding ablation studies, and strengthening the evaluation protocol with robustness checks. While some results remain strong relative to real-world benchmarks and the evaluation could be further expanded, the provided evidence is sufficiently clear and convincing to support the stated claims.

---

> ### Author Response · Authors · 2026-02-24
> **Camera-ready submission**
>
> Thank you. We have submitted the camera-ready version addressing all requested minor revisions.